# Assessing risk factors for latent and active tuberculosis among persons living with HIV in Florida: A comparison of self-reports and medical records

**Nana Ayegua Hagan Seneadza**[1]*, **Awewura Kwara**[2], **Michael Lauzardo**[2], **Cindy Prins**[3], **Zhi Zhou**[3], **Marie Nancy Séraphin**[2], **Nicole Ennis**[4], **Jamie P. Morano**[5], **Babette Brumback**[6], **Robert L. Cook**[3]

1 Department of Community Health, University of Ghana Medical School, Accra, Ghana, 2 Division of Infectious Diseases and Global Medicine, College of Medicine, University of Florida, Gainesville, Florida, United States of America, 3 Department of Epidemiology, College of Public Health and Health Professions and College of Medicine, University of Florida, Gainesville, Florida, United States of America, 4 Department of Behavioral Sciences and Social Medicine, Florida State University, Tallahassee, Florida, United States of America, 5 University of South Florida, Morsani College of Medicine, Tampa, Florida, United States of America, 6 Department of Biostatistics, Colleges of Public Health & Health Professions and Medicine, University of Florida, Gainesville, Florida, United States of America

* nseneadza@ug.edu.gh

**Data Availability Statement:** The data contain potentially sensitive patient information, but data can be obtained upon request. Information about

## Abstract

### Purpose

This study examined factors associated with TB among persons living with HIV (PLWH) in Florida and the agreement between self-reported and medically documented history of tuberculosis (TB) in assessing the risk factors.

### Methods

Self-reported and medically documented data of 655 PLWH in Florida were analyzed. Data on sociodemographic factors such as age, race/ethnicity, place of birth, current marital status, education, employment, homelessness in the past year and 'ever been jailed' and behavioural factors such as excessive alcohol use, marijuana, injection drug use (IDU), substance and current cigarette use were obtained. Health status information such as health insurance status, adherence to HIV antiretroviral therapy (ART), most recent CD4 count, HIV viral load and comorbid conditions were also obtained. The associations between these selected factors with self-reported TB and medically documented TB diagnosis were compared using Chi-square and logistic regression analyses. Additionally, the agreement between self-reports and medical records was assessed.

### Results

TB prevalence according to self-reports and medical records was 16.6% and 7.5% respectively. Being age ≥55 years, African American and homeless in the past 12 months were statistically significantly associated with self-reported TB, while being African American

the process to request and receive data from the Florida Cohort study are available from the Southern HIV and Alcohol Research Consortium (SHARC) at https://sharc-research.org/research/data/sharc-concepts-system/.

**Funding:** The Florida Cohort study was supported by The National Institute on Alcohol Abuse and Alcoholism (NIAAA), Grants U24AA022002 and U24AA022003. NAHS was supported under the University of Florida-University of Ghana Training Program in Tuberculosis and HIV Research in Ghana, funded by the Fogarty International Center at the National Institutes of Health, Grant TW010055.

**Competing interests:** The authors have declared that no competing interests exist.

**Abbreviations:** AIDS, Acquired immunodeficiency syndrome; ART, Anti-retroviral therapy; CD4, T lymphocyte cells; CI, Confidence interval; FDOH, Florida Department of Health; HIV, Human immunodeficiency virus; LTBI, Latent TB infection; NPV, Negative predictive value; OR, Odds ratio; PLWH, Persons living with HIV; PPD, Purified protein derivation; PPV, Positive predictive value; TB, Tuberculosis; US, United States.

homeless in the past 12 months and not on antiretroviral therapy (ART) were statistically significantly associated with medically documented TB. African Americans compared to Whites had odds ratios of 3.04 and 4.89 for self-reported and medically documented TB, respectively. There was moderate agreement between self-reported and medically documented TB (Kappa = 0.41).

## Conclusions

TB prevalence was higher based on self-reports than medical records. There was moderate agreement between the two data sources, showing the importance of self-reports. Establishing the true prevalence of TB and associated risk factors in PLWH for developing policies may therefore require the use of self-reports and confirmation by screening tests, clinical signs and/or microbiologic data.

## Introduction

The state of Florida carries a high burden of HIV, hosting an estimated 12% of all new HIV cases in the United States (US) in 2018 [1]. Similarly, Florida carries a high burden of active tuberculosis (TB), which since 2007 has been the leading infectious cause of death in persons living with HIV/AIDS (PLWH) globally [2]. In 2018, 591 (6.5%) of all TB cases in the US [3] were reported in Florida representing a 7.6% increase from 2017 when 549 new cases were reported in the state, whilst the rate of TB/HIV coinfection of 9% [4] exceeded the national rate of 5.3% [5].

Generally, PLWH, especially with more profound immunosuppression, are more at risk for developing active TB sometimes as a progression from latent TB infection (LTBI). This active TB accelerates morbidity and mortality in untreated HIV disease [6].

The degree to which different data sources provide similar prevalence estimates for TB risk factors in the same population of PLWH is unclear. Additionally, though factors such as race/ethnicity, age, sex, excessive alcohol use, smoking, homelessness, incarceration, and diabetes mellitus have been found to be associated with TB [5, 7, 8], only a few studies have looked at the association between TB and substance use in the US [8–10]. Currently, no studies have compared self-reports and medical records to examine the association between TB and substance use among PLWH in Florida. Knowing the relationship between the use of substances such as marijuana and smoked crack cocaine and TB infection is important in developing targeted prevention strategies for the comprehensive care of PLWH.

While self-reports are relatively easy and inexpensive to obtain, they may be limited by recall and/or social desirability bias, as well as by inconsistent responses depending on how questions are understood, and/or underreporting/overreporting depending on how measures are assessed [11, 12]. Medical records, on the other hand, are easily accessible by trained personnel, and data can be abstracted multiple times [13] However, medical records derived from routine care may not contain all the information relevant to a researcher and can be costly to obtain. Further, there may be differences in the extraction process and content, especially when different sites are involved [14, 15]. Studies have found varying agreement between self-reports and medical records, with chronic diseases and diseases with easily distinguishable diagnostic criteria having a higher agreement compared to acute conditions [16–18].

Thus, in this study, we measured the association between Latent TB Infection (LTBI)/TB and sociodemographic, behavioral, and health status factors and examined the agreement

between self-reports and medical records in assessing risk factors for LTBI/TB among PLWH in Florida.

## Methods

### Study design and population

This is a secondary analysis of baseline survey data from a Florida HIV cohort study. The survey was conducted from 2014 to 2018. Study participants included PLWH accessing healthcare at county health departments and community clinics in Florida. The survey collected information about sociodemographic characteristics lifestyle factors, comorbid conditions, and health outcomes associated with HIV in adults. Medical records of participants were also abstracted during the period of the survey. Details of the Florida Cohort study procedures have been previously described [19, 20].

Data included for analysis was on 655 individuals, 18 years and older, who responded to the survey question, "Have you ever been diagnosed with TB, or been told you have a positive skin test (sometimes called a PPD) or a positive TB blood test (called a Quantiferon Gold or T-spot test)?," and had medical records.

The research protocol was approved by the institutional review boards (IRBs) of the University of Florida (IRB201500849), Florida International University, and the Florida Department of Health. All participants provided written informed consent before participating in the study.

### Measures

**Outcome variables.** Self-reported (latent or active) TB was categorized based on the survey question above into 'yes' or 'no'. Medically documented TB diagnosis was also categorized into a 'yes' or 'no' binary outcome based on whether participants had any form of TB (latent or active) documented in their medical records using the International Classification of Diseases ICD-9 codes (010–017), 795.51, 795.52 or 10 codes (A15-A19), R76.11, R76.12, Z22.7.

**Sociodemographic variables.** Self-reported sociodemographic factors such as age, race/ethnicity, place of birth, current marital status, education, employment, homelessness in the past year and 'ever been jailed' were examined.

**Behavioral variables.** Excessive alcohol use was based on whether participants were consuming > 7 alcoholic drinks/week for women or > 14 alcoholic drinks/week for men [19]. Other variables included were 'ever used marijuana at least once weekly,' injection drug use (IDU) in the past 12 months, non-injection crack cocaine use, non-injection ecstasy use, injection stimulant use, and current cigarette use.

**Health status variables.** Based on their health insurance status, participants were categorized as 'insured' or 'uninsured'.

Adherence to HIV antiretroviral therapy (ART) was categorized into '≥ 95%' and '< 95%' based on the proportion of the last 30 days that the treatment was adhered to and 'Not on ART.'

The most recent CD4 count, HIV viral load and comorbid conditions (hepatitis C status and diabetes mellitus) obtained from the medical records were included in the analysis.

### Statistical analyses

Demographic characteristics and risk factors for TB were similar for individuals included and excluded from the study. The proportions of the total sample (prevalence) who had self-reported TB and medically documented TB was computed. The associations between the

factors listed and self-reported TB or medically documented TB were determined using Chi-square tests for categorical variables. Multivariable logistic regression analyses were conducted using either self-reported TB or medically documented TB as the outcome variable. In each of the two models, factors known to be associated with TB from literature with p-value $\leq$ 0.1 in the Chi-square or Fischer's exact test analysis were included to allow for the consideration of important factors which may not have shown significant association at p-value <0.05. Results of the logistic regression models are presented as odds ratios (OR) and 95% confidence intervals (CI). Sensitivity, specificity, positive predictive value (PPV), and negative predictive value (NPV) of self- reported TB compared to medically documented TB were constructed. The Kappa ($\kappa$) coefficient was used to determine the strength of agreement between self-reported TB and medically documented TB. Kappa coefficients were classified based on the following: $\leq$ 0.40 as no to fair agreement, 0.41–0.60 as moderate, 0.61–0.80 as substantial, and 0.81–1.00 as almost perfect agreement [16, 21]. We conducted complete case analysis, using the statistical software SAS 9.4 (SAS Institute, Cary NC). Significance level was set at p-value < 0.05.

## Results

Baseline characteristics of the 655 study participants are shown in Table 1. Majority of the participants were aged 45 years or above (59.2%), male (64.7%), African American (59.3%), US-born (86.9%), single/divorced/widowed/separated (80.3%), and unemployed (73.6%). Being homeless in the past 12 months was reported by 16.6%, while 65.2% reported ever being in jail.

Of the 655 participants, 109 (16.6%) had self-reported latent or active TB and 49 (7.5%) had medically documented TB. In general, among participants who had a particular risk factor assessed, a higher proportion had self-reported TB compared to medically documented TB (Table 2). Compared to younger individuals, older participants (45-54years and $\geq$ 55 years) were more likely to have self-reported TB but less likely to have medically documented TB. Being African American was more likely to be associated with both self-reported TB (22.0%) and medically documented TB (11.1%) than the other two categories of race (White or Other race/ethnicity) as shown in Table 2.

Participants who were US-born or who had less than a high school education were more likely to have either self-reported TB (16.6%, and 19.4% respectively) or medically documented TB (7.8% and 11.6% respectively) compared to other participants. Participants who had ever been in jail or who had been homeless in the past 12 months were more likely to have either self-reported TB (29.2% and 24.3% respectively) or medically documented TB (10.4% and 14.0% respectively). Age $\geq$ 45 years, being African American, incarceration, and homelessness were significantly associated with self-reported TB while being African American, less than high school education and homelessness were significantly associated with medically documented TB (p-value < 0.05).

Those who reported currently using non-injection crack cocaine were more likely to self-report TB (26.9%) or have medically documented TB in their (10.8%) compared to those who did not have these factors. Non-injection crack cocaine use was statistically significantly associated with self-reported TB, while not being on ART was significantly associated with having medically documented TB with p-value < 0.05 (Table 2). Other factors examined such as US born, marital status, employment, cigarette use, ever use of marijuana at least once weekly, injection drug use in the past 12 months, use of non-injection ecstasy and injection stimulants, CD4 count and hepatitis C infection were not significantly associated with either self-reported TB or TB based on medical records.

In the logistic regression model, being age $\geq$55 years compared to 18–34 years (OR = 2.79 95%CI; 1.23–6.30), African American compared to White (OR = 3.04, 95% CI; 1.65–5.59) and

**Table 1. Baseline characteristics of 655 persons living with HIV in the Florida Cohort who had medical records, 2014–2018.**

| Sociodemographic characteristics | Number (%) | Health status characteristics | Number (%) |
|---|---|---|---|
| **Age** | | **Health insurance** | |
| 18–34 | 127 (19.4) | Uninsured | 44 (7.0) |
| 35–44 | 140 (21.4) | Insured | 588 (93.0) |
| 45–54 | 249 (38.0) | | |
| ≥ 55 | 139 (21.2) | | |
| **Sex** | | **Adherence to ART** | |
| Male | 424 (64.7) | ≥ 95% | 382 (61.6) |
| Female | 231 (35.3) | < 95% | 175 (28.2) |
| | | Not on ART | 63 (10.2) |
| **Race/Ethnicity** | | **Viral load** | |
| White | 198 (30.3) | > 200 copies /ml | 137 (20.9) |
| African American | 387 (59.3) | ≤ 200 copies /ml | 518 (79.1) |
| Others | 68 (10.4) | | |
| **US Born** | | **Diabetes Mellitus** | |
| Yes | 565 (86.9) | Yes | 79 (12.1) |
| No | 85 (13.1) | No | 576 (87.9) |
| **Education** | | | |
| < High school | 216 (33.1) | | |
| High school diploma | 201 (30.8) | | |
| > High school | 236 (36.1) | | |
| **Ever been jailed** | | | |
| Yes | 412 (65.2) | | |
| No | 220 (34.8) | | |
| **Homeless** | | | |
| Yes | 107 (16.6) | | |
| No | 539 (83.4) | | |
| **Behavioral characteristics** | | | |
| **Cigarette use** | | | |
| Yes | 331 (47.6) | | |
| No | 301 (52.4) | | |
| **Excessive alcohol use (heavy drinking)** | | | |
| Yes | 57 (9.2) | | |
| No | 562 (90.8) | | |
| **Ever used marijuana on a regular basis-at least once per week** | | | |
| Yes | 358 (58.4) | | |
| No | 255 (41.6) | | |
| **Injection drug use in the past 12 months** | | | |
| Yes | 38 (6.1) | | |
| No | 585 (93.9) | | |
| **Non injection crack cocaine use** | | | |
| Never | 422 (67.2) | | |
| Past use—> 12months | 113 (18.0) | | |
| Current use—≤ 12 months | 93 (14.8) | | |

**Table 2. Factors associated with TB according to self-reports and medical records among 655 persons living with HIV in Florida.**

| Characteristic | Self-reported TB N (%) | P-value | TB based on medical records N (%) | P-value |
|---|---|---|---|---|
| **Sociodemographic Factors** | | | | |
| **Age (n)** | | | | |
| 18–34 (127) | 13 (10.2) | 0.01 | 9 (7.1) | 0.23 |
| 35–44 (140) | 16 (11.4) | | 6 (4.3) | |
| 45–54 (249) | 49 (19.7) | | 19 (7.6) | |
| ≥55 (139) | 31 (22.3) | | 15 (10.8) | |
| **Race/Ethnicity (n)** | | | | |
| White (198) | 17 (8.6) | <0.01 | 5 (2.5) | <0.01 |
| African American (387) | 85 (22.0) | | 43 (11.1) | |
| Others (68) | 7 (10.3) | | 1 (1.5) | |
| **Homeless (n)** | | | | |
| Yes (107) | 26 (24.3) | 0.02 | 15 (14.0) | <0.01 |
| No (539) | 82 (15.2) | | 33 (6.1) | |
| **Education (n)** | | | | |
| < High school (216) | 42 (19.4) | 0.12 | 25 (11.6) | 0.01 |
| High school diploma (201) | 37 (18.4) | | 15 (7.5) | |
| > High school (236) | 30 (12.7) | | 9 (3.8) | |
| **Sexual orientation (n)** | | | | |
| Heterosexual or straight (340) | 63 (18.5) | 0.11 | 27 (7.9) | 0.18 |
| Gay/lesbian (216) | 26 (12.0) | | 17 (7.9) | |
| Bisexual (64) | 12 (18.8) | | 1 (1.6) | |
| **Behavioral factors** | | | | |
| **Non injection crack cocaine use (n)** | | | | |
| Never (422) | 56 (13.3) | <0.01 | 30 (7.1) | 0.41 |
| Past use—>12months (113) | 25 (22.1) | | 7 (6.2) | |
| Current use—≤ 12 months (93) | 25 (26.9) | | 10 (10.8) | |
| **Health status factors** | | | | |
| **Health insurance (n)** | | | | |
| Uninsured (44) | 3 (6.8) | 0.07 | 3 (6.8) | 1.00** |
| Insured (588) | 101 (17.2) | | 43 (7.3) | |
| **Adherence to ART (n)** | | | | |
| ≥ 95% (382) | 63 (16.5) | 0.97 | 27 (7.1) | <0.01 |
| < 95% (175) | 30 (17.1) | | 10 (5.7) | |
| Not on ART (63) | 11 (17.5) | | 12 (24.5) | |
| **Viral load** | | | | |
| > 200 copies /ml (137) | 24 (17.5) | 0.76 | 9 (6.6) | 0.65 |
| ≤ 200 copies /ml (518) | 85 (16.4) | | 40 (7.7) | |
| **Diabetes Mellitus (n)** | | | | |
| Yes (79) | 18 (22.8) | 0.12 | 9 (11.4) | 0.16 |
| No (576) | 91 (15.8) | | 40 (6.9) | |

homeless in the past 12 months (OR = 1.39, 95% CI; 0.78–2.47), were statistically significantly associated with self-reported TB, while being African American (OR = 4.89, 95% CI; 1.67–14.36), homeless in the past 12 months (OR = 3.00, 95% CI; 1.46–6.15), 95% CI:), and not on ART(OR = 3.01, 95% CI; 1.31–6.91), 95% CI:), were statistically significantly associated with medically documented TB after adjusting for the other covariates (Table 3).

**Table 3. Multivariable logistic regression analysis of the association between risk factors in persons living with HIV in Florida and data sources on TB diagnosis.**

| Characteristic | Self-reported TB (N = 581) Adjusted Odds ratios (95% CI) | TB based on medical records (N = 570) Adjusted Odds ratios (95% CI) |
|---|---|---|
| **Age** | | |
| 18–34 | reference | |
| 35–44 | 1.03 (0.43–2.46) | |
| 45–54 | 1.95 (0.91–4.17) | |
| ≥55 | 2.79 (1.23–6.30) | |
| **Race/Ethnicity** | | |
| White | reference | reference |
| African American | 3.04 (1.65–5.59) | 4.89 (1.67–14.36) |
| Others | 1.46 (0.53–4.06) | 0.84 (0.09–7.83) |
| **Homeless in the past 12months** | | |
| No | reference | reference |
| Yes | 1.39 (0.78–2.47) | 3.00 (1.46–6.15) |
| **Education** | | |
| High School diploma or equivalent | reference | |
| <High School | 1.00 (0.57–1.74) | 1.79 (0.82–3.91) |
| >High School | 0.86 (0.47–1.55) | 0.81 (0.33–2.04) |
| **Sex orientation** | | |
| Heterosexual or straight | reference | |
| Gay or lesbian | 0.97 (0.55–1.71) | |
| Bisexual | 1.24 (0.59–2.63) | |
| **Non injection crack cocaine use** | | |
| Never | reference | |
| Past use (>12months) | 1.73(0.96–3.10) | |
| Current use (≤ 12 months) | 1.45 (0.75–2.77) | |
| **Health Insurance** | | |
| Yes | reference | |
| No | 0.31 (0.07–1.38) | |
| **Adherence** | | |
| > = 95% | | reference |
| <95% | | 0.74 (0.32–1.66) |
| Not on ART | | 3.01 (1.31–6.91) |
| **Diabetes Mellitus** | | |
| No | reference | |
| Yes | 1.23 (0.63–2.39) | |

Depending on the data source being used as reference, the sensitivity, specificity, positive and negative predictive values varied. When self-reported TB was compared to medically documented TB, these measures with their 95% CI were: sensitivity = 0.76 (95% CI = 0.61–0.87), specificity = 0.88 (95% CI = 0.85–0.91), positive predictive value = 0.34 (95% CI = 0.25–0.44), and negative predictive value = 0.98 (95% CI = 0.96–0.99) (Table 4). When medical records were compared to self-reports, the measures were as seen in Table 5. The Cohen's Kappa

**Table 4. Sensitivity, specificity, positive and negative predictive values and agreement of self-reported TB compared to TB based on medical records of PLWH in Florida.**

| Self-reported TB | TB diagnosis in medical records | | | | | | |
|---|---|---|---|---|---|---|---|
| | No | Yes | Sensitivity (95% CI) | Specificity (95% CI) | Positive Predictive Value (95% CI) | Negative Predictive Value (95% CI) | Cohen's Kappa (95% CI) |
| No | 534 | 12 | 0.76 (0.61–0.87) | 0.88 (0.85–0.91) | 0.34 (0.25–0.44) | 0.98 (0.96–0.99) | 0.41 (0.31–0.51) |
| Yes | 72 | 37 | | | | | |

statistic was 0.41 (95% CI = 0.31–0.51) showing moderate agreement between self-reported TB and medically documented TB (Tables 4 and 5).

## Discussion

We examined the prevalence of TB by self-reports and medical records, and the association between patient factors and TB based on self-reports and medical records and the agreement between the two data sources, The prevalence of self-reported TB (16.6%) exceeded medically documented TB (7.5%). Being African American was statistically significantly associated with both data sources on the history of TB. There was a moderate agreement between the two data sources on TB status.

In addition to the higher prevalence of self-reported TB compared to medically documented TB, the prevalence rates based on the two sources exceeded the estimate of 4.2% LTBI in PLWH [22] and TB/HIV coinfection rate of 5.3% in the US [5]. The self-reported TB rate of 16.6% in PLWH in Florida exceeded the 9% documented by the Florida Department of Health (FDOH) in 2018 [4] while the rate of 7.5% in medical records was slightly lower than the FDOH rate. The survey did not distinguish between LTBI and TB disease. This may explain the higher prevalence of self-reported TB compared to medically documented TB and make it difficult to compare the sources with respect to LTBI or TB disease prevalence. Persons diagnosed with LTBI or TB disease in other states or countries prior to HIV diagnosis may not have documentation in their medical records. Overreporting in self-reports and/or underreporting in medical records could have also resulted in the discrepancy in TB prevalence by data source. Patients who received testing, especially for latent TB, may report having been diagnosed with TB, especially if the medical evaluation was begun but not completed, or if the communication between the patient and provider was inadequate. Our findings suggest that among PLWH in care, self-reports may overestimate TB infection or disease prevalence while medical records of LTBI and active TB are incomplete and may lead to underreporting.

The factors associated with TB were generally similar (though not always statistically significantly) for both data sources although the proportions were higher in those who self-reported TB than in those who had medically documented TB. Being African American was significantly associated with TB based on both data sources. This finding is consistent with reported

**Table 5. Sensitivity, specificity, positive and negative predictive values and agreement of TB based on medical records compared to self-reported TB of PLWH in Florida.**

| TB diagnosis in medical records | Self-reported TB | | | | | | |
|---|---|---|---|---|---|---|---|
| | No | Yes | Sensitivity (95% CI) | Specificity (95% CI) | Positive Predictive Value (95%CI) | Negative Predictive Value (95%CI) | Cohen's Kappa (95% CI) |
| No | 534 | 72 | 0.34 (0.25–0.44) | 0.98 (0.96–0.99) | 0.76 (0.61–0.87) | 0.88 (0.85–0.91) | 0.41 (0.31–0.51) |
| Yes | 12 | 37 | | | | | |

risk factors for TB in the US [23]. African Americans and other racial minority populations in the US have been documented to be disproportionately affected by TB because of the higher prevalence of LTBI in these populations, especially among those who are non-US born [24]. However, in this study, the majority (86.9%) of participants were US-born. Homelessness is reported to be associated with an increased risk of LTBI [5] and TB disease [25–27]. For factors such as 'ever been jailed', 'cigarette use', 'non-injection crack cocaine use' and 'non-injection ecstasy use' that didn't show consistent associations with TB status in either data source, persons with these factors had higher proportions of TB compared to those who did not have these factors, irrespective of the data source.

In our study, two different logistic regression models were created for the association between factors and the outcome variables since the factors showing significant associations with the self-reported TB or medically documented TB were different. This explains why the factors that remained significant after controlling for the other variables differed. In the models, reporting African American was the only factor significantly associated with TB in both data sources while not being on ART and being homeless were significantly associated with TB from medical records. ART adherence is important to prevent virologic failure with emergence of drug resistance, HIV transmission or development of opportunistic infections, including TB. An adherence of 95% or more is required to improve immunity and outcomes in PLWH [28]. ART improves immune status, thereby reducing the risk of TB and TB deaths [29, 30].

The sensitivity, specificity and positive predictive values of self-reported TB compared to medical records were low compared to other studies that compared these two data sources on morbidity [13]. These results should, however, be interpreted with caution as none of these sources is the "gold standard." This study shows that if a participant self-reported no TB, there was a 98% chance that the medical records would also not have TB documented. Though the percent agreement between the two data sources was 87.2%, the overall agreement between the two data sources can be described as moderate based on the Kappa value of 0.41 [16, 21]. The lack of a strong agreement between the two sources could indicate that patients have firm recollections of experiences from previous conditions but little control of or access to the final information captured in their medical records.

Limitations of the study include the fact that the question used to assess self-reported TB failed to distinguish between latent and active TB, making it impossible to look at TB infection and disease groups separately. Recall bias could also have occurred as the participants may not accurately recall LTBI, especially if they were not treated. Further analysis comparing the types of medically documented TB showed that a higher proportion (86.2%) of those who had active TB also self-reported TB while only 60% of those with LTBI had self-reported TB. This further suggests that individuals with LTBI are less likely to self-report TB compared to those who had active disease either because they were not told, didn't remember, or were not treated so they didn't consider it important. Active TB, on the other hand, is symptomatic, often requiring at least 6 months of directly observed therapy, lending itself to stronger recall. Though the self-reports didn't state any time frame for the TB diagnosis, the medical records abstraction was based on the list of the participants' problems during their most recent visit to the health care provider. This could have introduced errors due to omission of information in the medical records during the visit. The analysis conducted also assumed that the absence of documentation of TB in the medical records meant the absence of either latent or active TB diagnosis, potentially introducing misclassification bias.

Despite the limitations, the strengths of this study are worth mentioning. This is the first study (to our knowledge) that has compared these two data sources on TB status in PLWH in Florida. The study allowed for the assessment of multiple factors potentially associated with TB

using the same sample of PLWH to give a better understanding of the similarities and differences in the association between the factors and a history of TB. Non-traditional risk factors such as marijuana and non-injection crack cocaine use were included to assess their association with TB, adding to existing knowledge about other potential factors associated with TB. Participants were recruited from diverse settings including county health departments, the private health sector and the community. Thus, the sample provides some insight into risk factors such as non-injection drug use which are not captured in the routine surveillance data in Florida.

## Conclusion

The prevalence of self-reported TB was higher (16.6%) than medically documented TB (7.5%). Being African American was significantly associated with TB status from both data sources. There was moderate agreement between the two data sources, showing the importance of self-reports. Establishing the true prevalence of TB and associated risk factors in PLWH for developing policies may therefore require the use of both self-reports and confirmation by screening tests, clinical signs and/or microbiologic data.

Future studies comparing these data sources with surveillance data in the TB registry in Florida as well as LTBI test results in those without active TB are necessary to determine the agreement between these sources of data using the same sample of PLWH. Non-traditional factors such as non-injection crack cocaine use could be further examined for their association with TB and considered during risk assessment during TB screening.

## Acknowledgments

The authors would like to thank the participants, the research teams and the participating sites in the Florida Cohort study, the team of the Southern HIV and Alcohol Research Consortium (SHARC) of the University of Florida and the University of Ghana Medical School. We would like to thank Li, Yancheng (Alex) for his support during data analysis and Dr. Carolyn Bradley for proof reading and editing the manuscript. We would like to thank coordinators of the University of Florida-University of Ghana Training Program in Tuberculosis and HIV Research in Ghana.

## Author Contributions

**Conceptualization:** Nana Ayegua Hagan Seneadza, Robert L. Cook.

**Data curation:** Nana Ayegua Hagan Seneadza, Zhi Zhou.

**Formal analysis:** Nana Ayegua Hagan Seneadza.

**Funding acquisition:** Robert L. Cook.

**Methodology:** Nana Ayegua Hagan Seneadza, Robert L. Cook.

**Supervision:** Awewura Kwara, Michael Lauzardo, Cindy Prins, Robert L. Cook.

**Writing – original draft:** Nana Ayegua Hagan Seneadza.

**Writing – review & editing:** Nana Ayegua Hagan Seneadza, Awewura Kwara, Michael Lauzardo, Cindy Prins, Zhi Zhou, Marie Nancy Séraphin, Nicole Ennis, Jamie P. Morano, Babette Brumback, Robert L. Cook.

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
