## [Decision Letter · Decision Letter 0]

9 May 2022

PONE-D-21-28142Assessing Risk Factors for Latent and Active Tuberculosis Among Persons Living with HIV in Florida: A Comparison of Self-Reports and Medical RecordsPLOS ONE

Dear Dr. Nana Ayegua Hagan Seneadza,

Thank you for submitting your manuscript to PLOS ONE. After careful consideration, we feel that it has merit but does not fully meet PLOS ONE’s publication criteria as it currently stands. Therefore, we invite you to submit a revised version of the manuscript that addresses the points raised during the review process.

We look forward to receiving your revised manuscript.

Kind regards,

Wenping Gong, Ph.D.

Academic Editor

PLOS ONE

Journal Requirements:

3. Thank you for stating the following financial disclosure: "RLC reports grants from NIH, during the conduct of the study. The Florida Cohort study was funded by The National Institute on Alcohol Abuse and Alcoholism (NIAAA) Grant U24 AA022002."

4. Thank you for stating the following in the Acknowledgments Section of your manuscript: "The authors would like to thank the participants, the research teams and the participating sites in the Florida Cohort study, the team of the Southern HIV and Alcohol Research Consortium (SHARC) of the University of Florida and the University of Ghana Medical School. We would like to thank Li, Yancheng (Alex) for his support during data analysis and Dr. Carolyn Bradley for proof reading and editing the manuscript. NAHS was supported by University of Florida-University of Ghana Training Program in Tuberculosis and HIV Research in Ghana funded by Fogarty International Center at the National Institutes of Health [grant number TW010055]. The Florida Cohort study was funded by NIAAA Grant U24 AA022002."

Please remove any funding-related text from the manuscript and let us know how you would like to update your Funding Statement. Currently, your Funding Statement reads as follows: "RLC reports grants from NIH, during the conduct of the study. The Florida Cohort study was funded by The National Institute on Alcohol Abuse and Alcoholism (NIAAA) Grant U24 AA022002."

Reviewers' comments:

Reviewer's Responses to Questions

**Comments to the Author**

1. Is the manuscript technically sound, and do the data support the conclusions?

Reviewer #1: Yes

2. Has the statistical analysis been performed appropriately and rigorously? 

Reviewer #1: Yes

3. Have the authors made all data underlying the findings in their manuscript fully available?

Reviewer #1: Yes

4. Is the manuscript presented in an intelligible fashion and written in standard English?

Reviewer #1: Yes

5. Review Comments to the Author

Reviewer #1: Review: Assessing Risk Factors for Latent and Active Tuberculosis Among Persons Living with

99 HIV in Florida: A Comparison of Self-Reports and Medical Record

Thank you for an interesting manuscript. I think it can benefit from the following edits.

Based on your abstract, “This study examined factors associated with TB among persons living with HIV (PLWH) in Florida”, however this is not mentioned in the methods, results, or conclusion sections of the abstract.

1. Abstract method: Include the factors associated with risk of TB that were collected.

2. Abstract results: Include factors that are associated with TB

3. The conclusion of the abstract does not seem to relate to the findings of the study “establishing the true prevalence of TB in PLWH for developing policies would require confirmation by screening tests, clinical signs and/or microbiologic data.”. Was this not done as part of the medical documentation of a history of TB?

Main Manuscript

Methods section: Reference number for the IRB approval

Lines 191-194: Repetition – on comorbid conditions, consider revising

“The most recent CD4 count, HIV viral load and comorbid conditions (hepatitis C status and 192 diabetes mellitus) obtained from the medical records were included in the analysis. 193 Comorbid conditions included in the analysis were hepatitis C and diabetes mellitus, based on 194 information from the medical records of the participants”

Medical documentation of TB: Please elaborate what does that mean, I assume based on the ICD usage it was a diagnosis of TB. Was it both latent and active TB?

Lines 242-243: “There were no statistically significant associations between the other factors 243 and self-reported TB, or medically documented TB (Table 2)” seems to be a repetition of what is highlighted in lines 248 – 252

“Other factors examined such as US born, marital 249 status, employment, cigarette use, ever use of marijuana at least once weekly, injection drug use 250 in the past 12 months, use of non-injection ecstasy and injection stimulants, CD4 count, and 251 hepatitis C infection were not significantly associated with either self-reported TB or TB based 252 on medical records”

6. PLOS authors have the option to publish the peer review history of their article (what does this mean?). If published, this will include your full peer review and any attached files.

Reviewer #1: **Yes: **Limakatso Lebina

---

## [Author Response · Author response to Decision Letter 0]

6 Jul 2022

Response to Academic Editor and Reviewers

The authors would like to thank the Academic Editor and Reviewers for the very constructive review comments provided for us to improve on our manuscript and submission to PLOS One. 

Journal Requirements:

Response: The title page, abstract pages and the body of the manuscript have been updated in compliance with PLOS One Style requirements.

Response: This is well-noted, and all funding information has been provided. 

The Florida Cohort study itself was funded by The National Institute on Alcohol Abuse and Alcoholism (NIAAA) Grant U24 AA022002. However, NAHS (the first and corresponding author) was supported under the University of Florida-University of Ghana Training Program in Tuberculosis and HIV Research in Ghana, which was funded by Fogarty International Center at the National Institutes of Health [grant number TW010055], to undertake a training programme at the University of Florida during the time of this assessment. 

3. Thank you for stating the following financial disclosure: "RLC reports grants from NIH, during the conduct of the study. The Florida Cohort study was funded by The National Institute on Alcohol Abuse and Alcoholism (NIAAA) Grant U24 AA022002."

Response: "The funders had no role in study design, data collection and analysis, decision to publish, or preparation of the manuscript." This statement is correct.

4. Thank you for stating the following in the Acknowledgments Section of your manuscript: "The authors would like to thank the participants, the research teams and the participating sites in the Florida Cohort study, the team of the Southern HIV and Alcohol Research Consortium (SHARC) of the University of Florida and the University of Ghana Medical School. We would like to thank Li, Yancheng (Alex) for his support during data analysis and Dr. Carolyn Bradley for proof reading and editing the manuscript. NAHS was supported by University of Florida-University of Ghana Training Program in Tuberculosis and HIV Research in Ghana funded by Fogarty International Center at the National Institutes of Health [grant number TW010055]. The Florida Cohort study was funded by NIAAA Grant U24 AA022002."

Please remove any funding-related text from the manuscript and let us know how you would like to update your Funding Statement. Currently, your Funding Statement reads as follows: "RLC reports grants from NIH, during the conduct of the study. The Florida Cohort study was funded by The National Institute on Alcohol Abuse and Alcoholism (NIAAA) Grant U24 AA022002."

Response: Thank you for this observation. All funding-related text has been removed from the section on Acknowledgement and the manuscript. Since NAHS was on a training grant at the University of Florida at the time of this work, the updated Funding statement should read as follows:

“RLC reports grants from NIH, during the conduct of the study. The Florida Cohort study was funded by The National Institute on Alcohol Abuse and Alcoholism (NIAAA) Grant U24 AA022002.” 

 “NAHS was supported under the University of Florida-University of Ghana Training Program in Tuberculosis and HIV Research in Ghana, which was funded by Fogarty International Center at the National Institutes of Health [Grant recipient was AW and grant number TW010055], to undertake a training programme at the University of Florida”. 

Response: Data cannot be shared publicly because it contains sensitive data on persons living with HIV in Florida. Data are available from the Southern HIV and Alcohol Research Consortium (SHARC), University of Florida, for researchers who meet the criteria for access to confidential data.

Reviewers' comments:

Reviewer's Responses to Questions

Comments to the Author

1. Is the manuscript technically sound, and do the data support the conclusions?

Reviewer #1: Yes

Response: Thank you for this conclusion

2. Has the statistical analysis been performed appropriately and rigorously?

Reviewer #1: Yes

Response: Thank you very much for agreeing that our analysis is appropriate and rigorous

3. Have the authors made all data underlying the findings in their manuscript fully available?

Reviewer #1: Yes

Response: Thank you 

4. Is the manuscript presented in an intelligible fashion and written in standard English?

Reviewer #1: Yes

Response: Thank you very much for agreeing that our presented in an intelligible fashion and written in standard English

5. Review Comments to the Author

Reviewer #1: Review: Assessing Risk Factors for Latent and Active Tuberculosis Among Persons Living with

99 HIV in Florida: A Comparison of Self-Reports and Medical Record

Thank you for an interesting manuscript. I think it can benefit from the following edits.

Based on your abstract, “This study examined factors associated with TB among persons living with HIV (PLWH) in Florida”, however this is not mentioned in the methods, results, or conclusion sections of the abstract.

Response: We appreciate this comment and agree that factors should be included in the abstract. 

1. Abstract method: Include the factors associated with risk of TB that were collected.

Response: This methods section of the abstract has been updated as follows: 

‘Data on sociodemographic factors such as age, race/ethnicity, place of birth, current marital status, education, employment, homelessness in the past year and ‘ever been jailed’ and behavioural factors such as excessive alcohol use, marijuana, injection drug use (IDU), substance and current cigarette use were obtained. Health status information such as health insurance status, adherence to HIV antiretroviral therapy (ART), most recent CD4 count, HIV viral load and comorbid conditions were also obtained.’ (Lines 107-113 of the revised manuscript with track changes)

2. Abstract results: Include factors that are associated with TB

Response: Thank you. The statement below ahs been included in the abstract.

‘Being age ≥55 years, African American and homeless in the past 12 months were statistically significantly associated with self-reported TB, while being African American homeless in the past 12 months and not on antiretroviral therapy (ART) were statistically significantly associated with medically documented TB.’ (Lines 119-122 of the revised manuscript with track changes)

3. The conclusion of the abstract does not seem to relate to the findings of the study “establishing the true prevalence of TB in PLWH for developing policies would require confirmation by screening tests, clinical signs and/or microbiologic data.”. Was this not done as part of the medical documentation of a history of TB?

Response: Thank you for this. We agree that this conclusion does not seem to relate to the findings of the study. We have modified the phrase accordingly to read as follows:

‘There was moderate agreement between the two data sources, showing the importance of self-reports. Establishing the true prevalence of TB and associated risk factors in PLWH for developing policies may therefore require the use of both self-reports and confirmation by screening tests, clinical signs and/or microbiologic data.’ (Lines 126-130 of the revised manuscript with track changes) 

This statement has also been included in the conclusion of the main manuscript.

Main Manuscript

Methods section: Reference number for the IRB approval

Response: The reference number for the IRB approval (IRB201500849) has been included in the Methods section. (Line 184 of the revised manuscript with track changes)

Lines 191-194: Repetition – on comorbid conditions, consider revising

“The most recent CD4 count, HIV viral load and comorbid conditions (hepatitis C status and 192 diabetes mellitus) obtained from the medical records were included in the analysis. 193 Comorbid conditions included in the analysis were hepatitis C and diabetes mellitus, based on 194 information from the medical records of the participants”

Response: Thank you for pointing out the repetition in these statements. The statement “Comorbid conditions included in the analysis were hepatitis C and diabetes mellitus, based on information from the medical records of the participants” has been deleted. 

Medical documentation of TB: Please elaborate what does that mean, I assume based on the ICD usage it was a diagnosis of TB. Was it both latent and active TB?

Response: Medical documentation of TB was for both latent and active TB based on the International Classification of Diseases ICD-9 codes (010-017), 795.51, 795.52 or 10 codes (A15-A19), R76.11, R76.12, Z22.7 and this has been captured under the description of the outcome variables. (Lines 190-193 of the revised manuscript with track changes). 

Lines 242-243: “There were no statistically significant associations between the other factors 243 and self-reported TB, or medically documented TB (Table 2)” seems to be a repetition of what is highlighted in lines 248 – 252

“Other factors examined such as US born, marital 249 status, employment, cigarette use, ever use of marijuana at least once weekly, injection drug use 250 in the past 12 months, use of non-injection ecstasy and injection stimulants, CD4 count, and 251 hepatitis C infection were not significantly associated with either self-reported TB or TB based 252 on medical records”

Response: Thank you for pointing out the repetition in these statements.

The statement “There were no statistically significant associations between the other factors and self-reported TB, or medically documented TB (Table 2)” has been deleted. 

6. PLOS authors have the option to publish the peer review history of their article (what does this mean?). If published, this will include your full peer review and any attached files.

Do you want your identity to be public for this peer review? For information about this choice, including consent withdrawal, please see our Privacy Policy.

Reviewer #1: Yes: Limakatso Lebina

---

## [Editor Report · Decision Letter 1]

11 Jul 2022

Assessing Risk Factors for Latent and Active Tuberculosis Among Persons Living with HIV in Florida: A Comparison of Self-Reports and Medical Records

PONE-D-21-28142R1

Dear Dr. Nana,

We’re pleased to inform you that your manuscript has been judged scientifically suitable for publication and will be formally accepted for publication once it meets all outstanding technical requirements.

Kind regards,

Wenping Gong, Ph.D.

Academic Editor

PLOS ONE

---

## [Editor Report · Acceptance letter]

26 Jul 2022

PONE-D-21-28142R1 

Assessing Risk Factors for Latent and Active Tuberculosis Among Persons Living with HIV in Florida: A Comparison of Self-Reports and Medical Records 

Dear Dr. Seneadza:

I'm pleased to inform you that your manuscript has been deemed suitable for publication in PLOS ONE. Congratulations! Your manuscript is now with our production department. 

Kind regards, 

on behalf of

Dr. Wenping Gong 

Academic Editor

PLOS ONE